# Molecular Dynamics Simulation Reveals the Mechanism of Substrate Recognition by Lignin-Degrading Enzymes

**DOI:** 10.3390/ijms26199378

**Published:** 2025-09-25

**Authors:** Xue Ma, Xueting Cao, Zhenyu Ma, Jingyi Zhu, Letian Yang, Min Xiao, Xukai Jiang

**Affiliations:** 1National Glycoengineering Research Center, Shandong University, Qingdao 266237, China; 2State Key Laboratory of Microbial Technology, Shandong University, Qingdao 266237, China

**Keywords:** ligninolytic enzymes, biodegradation mechanism, molecular dynamics simulation

## Abstract

Lignin, the most abundant aromatic biopolymer, represents a key renewable feedstock for sustainable biorefineries, yet its structural complexity poses a formidable challenge for enzymatic degradation. While ligninolytic enzymes such as laccases (LACs), lignin peroxidases (LiPs), and manganese peroxidases (MnPs) exhibit remarkable catalytic versatility, the molecular mechanisms underlying their ability to balance substrate specificity and structural flexibility remain unresolved. Here, we employed all-atom molecular dynamics (MD) simulations and virtual mutagenesis to dissect the dynamic interactions between these enzymes and lignin model compound (β-O-4-linked H-type dimers). Our simulations revealed a dual recognition mechanism in which polar residues (such as Asp, Glu, Arg and His) formed hydrogen bonds with hydroxyl and keto groups near catalytic cleavage sites, ensuring precise alignment for bond scission, while aromatic residues stabilized diverse lignin conformations via hydrophobic interactions with conserved aromatic rings. Conformational dynamics of active-site residues enabled adaptive adjustments to substrate heterogeneity, reconciling enzymatic specificity with structural promiscuity. Virtual mutation experiments further demonstrated that aromatic residues were indispensable for binding stability, whereas polar residues dictated cleavage-site selectivity. These findings provide atomic-scale insights into the catalytic mechanism of ligninolytic enzymes, with implications in the rational design of superior biocatalyst for lignin biorefineries.

## 1. Introduction

The transition toward sustainable biorefineries demands innovative strategies to valorize lignin, the second most abundant biopolymer and nature’s richest source of renewable aromatic carbon. Despite its immense potential, lignin’s recalcitrant heteropolymer structure—composed of cross-linked guaiacyl (G), syringyl (S), and p-hydroxyphenyl (H) units interconnected by diverse bonds (β-O-4, β-5, β-β, etc.)—has hindered its efficient conversion into value-added chemicals [1,2,3,4] (Figure 1). Conventional degradation methods, relying on harsh chemical pretreatments or energy-intensive physical processes, face critical limitations in scalability, cost, and environmental sustainability [5,6]. In contrast, enzymatic degradation offers an eco-friendly alternative, with ligninolytic oxidoreductases such as laccases (LACs), lignin peroxidases (LiPs), and manganese peroxidases (MnPs) demonstrating remarkable catalytic versatility [7,8,9].

In recent years, density functional theory (DFT) studies have provided atomistic insights into enzyme–lignin model compound interactions, revealing key roles of electronic structure and active-site configuration in facilitating radical formation and bond cleavage. For instance, DFT calculations have quantified bond dissociation enthalpies across a range of lignin linkages including β-O-4, highlighting the influence of substituents and structural motifs on bond strength [10]. Other work modeling specific β-O-4 dimers demonstrated that concerted mechanisms often dominate over homolytic pathways, with functional group effects on reaction preferences [11]. Extending to more polymer-representative systems, DFT models of oligomeric lignin (e.g., 10-unit syringyl chain) revealed position-dependent bond reactivities and detailed thermodynamic behaviors [12]. Collectively, these computational studies emphasize that electronic factors and precise active-site architecture co-determine catalytic efficiency across ligninolytic enzymes. However, the molecular principles governing their substrate recognition and catalytic adaptability remains poorly resolved, impeding the rational design of high-performance biocatalysts for industrial applications.

Central to this challenge is the paradoxical catalytic behavior of ligninolytic enzymes: they must achieve precise cleavage of target bonds (e.g., β-O-4 linkages) while accommodating the structural diversity inherent to lignin’s side-chain substituents [5,13,14]. For instance, methoxylation patterns, hydroxyl group positioning, and aromatic ring substitutions vary significantly across G-, S-, and H-type units, yet these enzymes maintain catalytic efficiency across substrates. While prior studies have established the role of redox-active sites in generating lignin radicals [15,16], the mechanisms enabling substrate discrimination and binding stability remain enigmatic. Do these enzymes employ conserved structural motifs to recognize universal lignin features, or do they rely on dynamic conformational adjustments to process chemically distinct substrates? Resolving this question is critical for bridging the gap between lignin’s structural complexity and enzymatic degradation efficiency.

Current understanding of lignin–enzyme interactions is largely derived from static crystal structures and biochemical assays [17,18,19,20,21], which fail to capture the dynamic interplay between enzymes and lignin’s conformationally flexible substrates. Molecular dynamics (MD) simulations have emerged as a powerful tool to probe these interactions at atomic resolution [22,23], yet systematic comparative analyses of LACs, LiPs, and MnPs are lacking. Previous computational studies have focused on individual enzymes or simplified substrates, overlooking the cooperative roles of hydrogen bonding, hydrophobic interactions, and residue dynamics in substrate recognition.

In this study, we integrated comparative MD simulations, binding free energy calculations, and virtual mutagenesis to dissect the substrate recognition mechanisms of three major ligninolytic enzyme classes. The results revealed that all three enzyme classes employed a dual recognition strategy: Polar residues formed specific hydrogen bonds with hydroxyl and keto groups near cleavage sites, ensuring precise alignment of catalytic centers, while aromatic residues engaged in hydrophobic interactions with lignin’s conserved aromatic rings, providing broad substrate tolerance. Notably, conformational flexibility of active-site residues enabled dynamic adjustments to accommodate the structural heterogeneity of lignin—a feature not apparent in static structural models. Virtual mutagenesis further confirmed that aromatic residues were indispensable for binding stability, whereas polar residues dictated cleavage-site specificity. These findings inform a unified mechanism model of lignin recognition, where hydrogen bonding and hydrophobic forces act synergistically to balance the enzymatic specificity and flexibility.

## 2. Results

### 2.1. Structural Basis of Ligand Recognition by Lignin-Degrading Enzymes

To investigate the catalytic mechanisms of lignin-degrading enzymes toward β-O-4-linked ligands, we first performed molecular docking of a dimeric model compound containing the β-O-4 linkage with laccase, lignin peroxidase, and manganese peroxidase, respectively (Figure 2(a_1_)), followed by static structural analysis of their active sites. In the local structural view, key aromatic and negatively charged residues involved in ligand interactions are represented in stick form (Figure 2(b_1_)). Taking laccase 1GYC as an example, the ligand primarily interacts with residues F162 and F265 through hydrophobic forces, forms hydrogen bonds with H395, H458, and D206, and establishes van der Waals contacts with I455 (Figure 2(c_1_)). Surface hydrophobicity and electrostatic potential mapping revealed that hydrophobic residues are predominantly located near the two aromatic ends of the ligand (Figure 2(d_1_)), while the region surrounding the central cleavage bond is enriched with electronegative amino acids (Figure 2(e_1_)). This spatial distribution of residues helps stabilize the ligand conformation and guides it toward the cleavage site, thereby facilitating catalysis. A similar amino acid distribution pattern was also observed in lignin peroxidase and manganese peroxidase (Figure 2), suggesting that this structural feature may represent a shared mechanism by which lignin-degrading enzymes recognize and bind to their substrates.

### 2.2. Dynamic Interaction Mechanisms in Specific Substrate Recognition

To further elucidate the molecular basis of substrate binding by lignin-degrading enzymes, we conducted statistics and analysis of various interactions within the enzyme–substrate complexes. Root-mean-square deviation (RMSD) calculations indicated that all enzyme–ligand complexes reached equilibrium during molecular dynamics simulations, confirming their suitability for subsequent analysis (Appendix A). Compared with the enzyme-only systems, the substrate-bound complexes exhibited lower backbone RMSD fluctuations and thus greater overall structural stability throughout the simulation (Figure 3a), confirming that the ligand contributes to stabilizing the active-site conformation. We then examined the interaction energies of residues around the ligand (Figure 3b), revealing that hydrophobic interactions were the predominant contributors, while electrostatic interactions played a relatively minor role. Remarkably, laccase, lignin peroxidase, and manganese peroxidase shared a consistent interaction pattern during ligand recognition (Appendix A). These results suggest that hydrophobic residues substantially enhance the structural stability of the enzyme-ligand complex through stable hydrophobic interactions.

Building on this, we calculated the binding free energies between enzymes and individual functional groups of the model substrate to evaluate the preferential recognition of specific part of ligand (Figure 3d). The results showed that the methylene group adjacent to the cleavage site exhibited the highest binding free energy, followed by the neighboring methylene group and the flanking aromatic rings. This trend was consistently observed across multiple laccases, lignin peroxidases, and manganese peroxidases, suggesting that these regions may serve as key anchoring points for substrate binding (Appendix A). Additionally, functional groups containing oxygen atoms generally showed less favorable binding free energies, which is likely attributable to electrostatic repulsion between the electronegative oxygen atoms in the ligand and negatively charged regions within the enzyme active site (Figure 2e).

To further reveal the specificity of enzyme-ligand interactions, we quantified the number of hydrogen bonds formed between enzymes and various functional groups of the substrate during molecular dynamics simulations. Notably, laccase 1GYC formed specific hydrogen bonds with the –OH group of the lignin model compound, and a few hydrogen bonds were also observed at the –O3 position (Figure 3e). Comparative analysis across different enzymes confirmed that –OH groups consistently served as critical interaction sites during substrate recognition. Taken together, our findings demonstrate that laccase, lignin peroxidase, and manganese peroxidase recognize the cleavage site of lignin model compounds with high precision by forming specific hydrogen bonds with key functional groups—particularly hydroxyl groups (Appendix A). These directional hydrogen-bonding interactions not only enhance the stability of the enzyme–ligand complex but also provide a structural basis for the catalytic cleavage of the substrate.

Moreover, structural trajectory analyses from MD simulations identified catalytically relevant residues and their conformational rearrangements (Figure 3c). During the simulation, residues F265 and F162 rotated by 5.2 Å and 5.6 Å, respectively, toward the aromatic rings of the lignin model compound, while D206 and H458 shifted by 2.3 Å and 1.7 Å, respectively, toward the groups near the cleavage site. These movements ensured stable interactions with the substrate. These results suggest that active-site residues in lignin-degrading enzymes undergo dynamic conformational adjustments to align with key functional groups of the ligand, thereby enhancing binding stability.

Specifically, functional groups near the cleavage site primarily formed hydrogen bonds and other stabilizing interactions with polar residues, whereas the conserved terminal aromatic rings mainly engaged in hydrophobic interactions with aromatic residues. These key residues underwent significant dynamic adjustments during the simulation to maintain stable interactions. This adaptive mechanism provides a molecular basis for the efficient recognition of diverse lignin units with different substitution patterns—such as guaiacyl (G-type), syringyl (S-type), and p-hydroxyphenyl (H-type)—thereby explaining the high catalytic efficiency of lignin-degrading enzymes toward structurally heterogeneous substrates.

### 2.3. Functional Dissection of Hydrophobic and Hydrogen-Bonding Interactions via Site-Directed Mutagenesis

To further verify the distinct functional roles of hydrophobic and hydrogen-bonding interactions during catalysis, we performed virtual mutagenesis on key aromatic and polar residues of three representative lignin-degrading enzymes: laccase 1GYC, lignin peroxidase 1B82, and manganese peroxidase 2BOQ. Molecular dynamics simulations were conducted for each mutant. Analysis of the RMSD values of the lignin model compound revealed a marked increase and greater fluctuation in the mutants Lac1GYC-F265A, LiP1B82-F267A, and MnP2BOQ-F91A throughout the simulation period (Figure 4a). This indicates that replacing aromatic residues with alanine significantly compromises the stability of enzyme-substrate binding.

In addition, binding free energy calculations showed that the enzyme-ligand interactions in these mutants were substantially weakened (Figure 4b). Specifically, the binding free energies of Lac1GYC-F265A, LiP1B82-F267A, and MnP2BOQ-F91A were significantly lower than those of their respective wild-type enzymes, further confirming that hydrophobic interactions play a critical role in stabilizing enzyme–substrate complexes. Taken together, these results demonstrate that in the catalytic degradation of lignin by laccase, lignin peroxidase, and manganese peroxidase, hydrogen-bonding interactions primarily confer specificity in substrate recognition, whereas hydrophobic interactions are essential for maintaining the structural stability of the bound lignin model compound.

### 2.4. Universality of Catalytic Mechanisms Across Diverse Lignin-Degrading Enzyme

To validate the universality of the precise catalytic mechanism of lignin degradation across different classes of ligninolytic enzymes, we performed molecular dynamics simulations on four representative enzyme–substrate complexes for each of laccases, lignin peroxidases, and manganese peroxidases. We statistically analyzed the physicochemical properties of key residues that contributed significantly to the binding free energy (Figure 5a), monitored the positional shifts in residues surrounding the substrate during MD simulations (Figure 5b), and quantitatively evaluated the fluctuations of these residues (Figure 5c).

The statistical analysis of residue types contributing to binding free energy revealed that hydrophobic amino acids accounted for the majority in nearly all cases (Figure 5a). In laccases, the proportion of hydrophobic residues among the key binding contributors was approximately 70% for Lac1GYC, Lac4JHU, and Lac5E9N, and as high as 94% for Lac1V10. A similar trend was observed for lignin peroxidases: hydrophobic residues made up over 50% in LiP7OO5 and LiP1B82, and nearly 80% in LiP6A6Q and LiP1LLP. Among manganese peroxidases, MnP2BOQ showed 100% hydrophobic contributors, while MnP4CZO and MnP8QX0 had over 70%, and MnP1YYD had a notably lower proportion of 38%.

During the MD simulations, we observed that, across all three classes of enzymes, aromatic residues (such as Phe, Tyr, and Trp) and charged polar residues (such as Asp, Glu, Arg, and His) in close proximity to the substrate exhibited dynamic swinging motions oriented toward the ligand (Figure 5b). These substrate-facing conformational adjustments not only enhanced the stability of enzyme–substrate binding but may also contribute to the precise recognition of the scissile bond. This dynamic trend was consistently observed in multiple laccases, lignin peroxidases, and manganese peroxidases. Quantitatively, each enzyme system contained at least one aromatic residue with a root mean square fluctuation (RMSF) greater than 2 Å, among which F267 in LiP1LLP displayed a fluctuation amplitude as high as 7.1 Å, suggesting a strong capacity for conformational adaptability.

In addition, several charged polar residues were not only spatially mobile but also exhibited clear conformational changes. For instance, in LiP7OO5, the carboxyl group of residue E163 progressively moved toward the hydroxyl group near the cleavage site of the substrate, forming a stable hydrogen-bonding network. This interaction suggests a potential role for E163 in stabilizing the substrate or facilitating transition state formation during catalysis.

To further assess the statistical relevance of the observed residue dynamics, we compared the RMSF values of key recognition/catalytic residues located within 5 Å of the substrate to those of neighboring non-functional residues (Figure 5c). The violin plot revealed a right-skewed distribution for the key residues, with significantly higher fluctuation amplitudes compared to adjacent non-critical residues. These results indicate that the key residues exhibit greater conformational flexibility, which may provide the necessary structural adaptability for substrate-induced fit and catalytic function.

### 2.5. Conservation of Catalytic Motifs in Lignin-Degrading Enzyme

To explore the general principles underlying lignin recognition and catalytic mechanisms in ligninolytic enzymes, we performed phylogenetic analysis (Figure 6a) and sequence conservation analysis (Figure 6b) for laccases, lignin peroxidases, and manganese peroxidases. The phylogenetic tree revealed that ligninolytic enzymes are divided into two major evolutionary branches: laccases form a relatively distinct lineage, whereas lignin peroxidases and manganese peroxidases exhibit higher sequence similarity and share a closer evolutionary relationship.

Further sequence conservation analysis demonstrated that most of the key residues at the catalytic centers are highly conserved among the three enzyme classes, while considerable sequence variability exists outside these regions (Figure 6b). In particular, polar residues involved in hydrogen bonding and aromatic residues responsible for hydrophobic interactions are strictly conserved within the active sites. This indicates that, beyond the four representative enzymes we analyzed, a broader range of ligninolytic enzymes share conserved structural motifs around their active centers. These conserved elements likely play a critical role in recognizing the interunit chemical linkages and aromatic ring structures characteristic of lignin substrates.

## 3. Discussion

In this study, we employed all-atom molecular dynamics simulations to systematically investigate the interactions between several representative laccases, lignin peroxidases, and manganese peroxidases with lignin model compounds, aiming to elucidate the molecular mechanisms underlying lignin substrate recognition by ligninolytic enzymes. Our results reveal that ligninolytic enzymes achieve precise cleavage of key interunit linkages (such as β-O-4 bonds) primarily through specific hydrogen bonds formed between polar amino acid residues and functional groups adjacent to the scissile bonds. This mechanism has also been experimentally evidenced in LiP, where the hydrogen bond between His-239 and Asp-238 plays a crucial role in lignin polymer binding [24]. Concurrently, the presence of aromatic residues enables nonspecific hydrophobic interactions with the lignin side chain substituents, accommodating the structural heterogeneity inherent to lignin, similar to synthetic polymers containing aromatic amino acids that bind lignin via side-chain hydrophobic and aromatic clustering [25]. These findings provide a theoretical framework for the rational design of ligninolytic enzymes to facilitate the industrial-scale degradation and valorization of lignin.

One of the most significant findings is the capacity of laccases, lignin peroxidases, and manganese peroxidases to specifically recognize functional groups located near the catalytic cleavage sites of the lignin substrates. MD simulations revealed that this recognition is mediated by hydrogen bonds formed between polar residues and the key groups flanking the cleavage site. Similar hydrogen bonding interactions have been reported in previous studies, where polar residues such as Asp, Glu, Arg, His, and Thr in lignin peroxidases formed hydrogen bonds with various lignin s substrates [26]. In laccases, hydrogen bonding has also been observed with 2,6-dimethoxyphenol, ferulic acid, and sinapic acid [27,28]. In LiP, structural studies further demonstrated that the hydrogen bond between His-82 and substrate hydroxyl groups plays a critical role in substrate positioning [29]. Unlike static analyses in prior studies, our dynamic simulations offer a more detailed mapping of specific hydrogen bonding interactions across different enzyme classes, thereby identifying key substrate features that may serve as potential targets for catalytic enhancement. Moreover, molecular docking and MD simulation studies have also shown that laccases interact with lignin model compounds through both hydrogen bonding and hydrophobic interactions, which strongly supports our findings [28].

Another important insight from our study is that hydrophobic interactions are critical for stabilizing the binding of lignin substrates with diverse side chain substituents. Structural analysis revealed that conserved aromatic residues surrounding the aromatic rings of lignin model compounds frequently engage in hydrophobic interactions. Site-directed mutagenesis confirmed the pivotal role of these interactions in maintaining substrate-binding stability. This observation is supported by earlier studies, where hydrophobic residues such as Phe, Tyr, Leu, and Val were found to interact consistently with lignin model compounds, though their spatial distribution varied among different enzyme types [26,30,31]. Notably, LiP structural analyses revealed that hydrophobic residues such as Leu-167 and Phe-267 surrounding the critical Trp-171 may contribute to lignin binding by enhancing hydrophobic environments [24]. Furthermore, it has been widely recognized that aromatic residues often act as anchoring elements at protein–ligand interfaces, highlighting their universal role in substrate recognition [32].

Taken together, our all-atom MD simulations provide a comprehensive mechanistic understanding of how ligninolytic enzymes achieve both specificity and versatility in substrate recognition. Specifically, these enzymes utilize hydrogen bonding to recognize conserved functional groups near catalytic cleavage sites, while hydrophobic interactions with aromatic substituents contribute to stable substrate binding despite structural diversity. This dual-interaction mechanism offers a new conceptual framework for engineering ligninolytic enzymes with improved catalytic performance. In addition, recent studies have demonstrated that targeted structural modifications can significantly enhance the functionality of versatile peroxidase enzymes, providing valuable inspiration for applying our framework to future enzyme engineering [33].

Beyond the molecular insights, the broader implications of this study extend to both social and economic domains. From an industrial perspective, improved enzyme design could significantly lower the costs associated with lignin depolymerization, enabling more efficient production of bio-based chemicals, fuels, and value-added aromatics. Such advances may reduce reliance on fossil resources, contributing to the development of a circular bioeconomy and more sustainable chemical industries. On a societal level, enhanced lignin valorization technologies could help mitigate environmental burdens associated with agricultural and forestry residues, reducing waste while generating renewable feedstocks. Furthermore, by fostering new economic opportunities in biomass utilization and green chemistry, these findings may support rural economies, promote job creation in bio-based industries, and advance global efforts toward carbon neutrality.

## 4. Materials and Methods

### 4.1. Protein Preparation

To explore the general principles in lignin-degrading enzymes, this study includes a dataset comprising homologous laccases from the AA1 family, as well as lignin peroxidases and manganese peroxidases from the AA2 family. Four laccases were selected: *Tv*lac (PDB: 1GYC) from *Trametes versicolor*, *Rm*L (PDB: 1V10) from *Rigidoporus microporus*, *Sm*L (PDB: 5E9N) from *Steccherinum murashkinskyi*, and *Cc*L (PDB: 4JHU) from *Cerrena caperata*. Four lignin peroxidases were chosen: *Ape*LiP (PDB: 7OO5) from *Agrocybe pediades* and *Pch*-LiPs (PDBs: 6A6Q, 1B82, 1LLP) from *Phanerodontia chrysosporium*. Additionally, four manganese peroxidases were selected: *Ape*-MnP (PDB: 8QX0) from *Agrocybe pediades*, *Gs*MnP (PDB: 4CZO) from *Gelatoporia subvermispora*, *Pe*MnP (PDB: 2BOQ) from *Pleurotus eryngii*, and *Pc*MnP (PDB: 1YYD) from *Phanerodontia chrysosporium*. All protein structures were retrieved from the Protein Data Bank (https://www.rcsb.org, accessed on 30 January 2025). These enzymes are distributed across different locations in the phylogenetic tree of the Auxiliary Activities (AA) families 1 and 2. Their structural similarities and functional variations make them ideal candidates for investigating the structure–catalysis relationships of AA lignin-degrading enzymes.

### 4.2. Molecular Dynamics Simulations

As a widely accepted form of theoretical derivation, all molecular dynamics (MD) simulations were performed using GROMACS 2019 software with the CHARMM36 all-atom force field. Proteins were solvated using the SPC model in a cubic box with dimensions of 8 × 8 × 8 nm^3^ [34]. To ensure a neutral system with an ionic concentration of 0.15 mol·L^−1^, Na^+^ and Cl^−^ ions were added by randomly replacing water molecules [35]. Steepest energy minimization was performed for 5000 steps for each system to reduce the maximum force to below 1000 kJ·mol^−1^·nm^−2^. The system was then equilibrated for 200 ps under the isothermal–isochoric (NVT) ensemble, followed by an additional 200 ps under the isothermal–isobaric (NPT) ensemble [36]. Equilibration was verified by the convergence of potential energy and temperature. A 100 ns production MD simulation was conducted with three replicas in the NPT ensemble, using the V-rescale and Parrinello–Rahman methods to control the temperature (303.15 K) and pressure (1 bar), respectively [37,38,39]. The LINCS algorithm was applied to constrain bonds involving hydrogen atoms, and the SETTLE algorithm was employed for water molecules [40,41]. Long-range electrostatic interactions were calculated using the Particle Mesh Ewald (PME) method [42]. The non-bonded pair list cutoff was set to 10.0 Å, with updates every 10 fs. Two MD systems were prepared for each lignin-degrading enzyme: a protein-only system and a protein–substrate complex system.

### 4.3. Data Analysis

In order to conduct a multi-level analysis of the results, the biophysical properties of lignin-degrading enzymes were analyzed using internal tools in GROMACS. A hydrogen bond was defined using the gmx_hbond tool if the acceptor–donor distance was less than 0.35 nm and the acceptor–hydrogen–donor angle was less than 30° [43]. Structural stability and flexibility were assessed by calculating the root mean square deviation (RMSD, gmx_rms). The Cα atoms of each snapshot structure were superimposed onto the initial structure using least-squares fitting [44]. The interaction energy between amino acid residues and substrates was calculated using the CHARMM36 all-atom force field (gmx_energy) [45]. The MMPBSA method (gmx_mmpbsa) was employed to calculate binding free energies for non-covalently bound complexes. Structural visualization was performed using PyMOL (Version 3.0 Schrödinger, LLC, New York City, NY, USA), while the receptor–ligand interactions were displayed by Discovery Studio Visualizer (version 24.0.1.232).

For docking studies, the GLIDE module of Schrödinger (Glide v, Schrödinger) was employed. Protein preparation and refinement were performed using the Protein Preparation Wizard in the Schrödinger Suite. The significance levels (*p*-values) in Figure 5c were obtained using the Mann–Whitney U test (two-sided). All analyses were performed in Python (version 3.1), using a custom script based on itertools for exact distribution enumeration. The evolutionary analysis was conducted using MEGA 11. A total of 84 sequences (40 laccases, 22 lignin peroxidases, and 22 manganese peroxidases) were retrieved from the NCBI database and aligned using Clustal Omega (https://www.ebi.ac.uk/jdispatcher/msa/clustalo?stype=protein, accessed on 30 January 2025). Sequence profiles for the active sites of lignin-degrading enzymes were generated using WebLogo (https://weblogo.berkeley.edu, accessed on 30 January 2025).

## 5. Conclusions

This study provides detailed molecular insights into how ligninolytic enzymes recognize and degrade lignin substrates. Through all-atom molecular dynamics simulations, we show that hydrogen bonding with conserved functional groups ensures catalytic specificity, while hydrophobic interactions with aromatic substituents stabilize substrate binding despite lignin’s heterogeneity. Together, these dual mechanisms explain the balance between specificity and versatility in lignin depolymerization. Our findings complement and extend previous structural and computational studies, offering a more dynamic view of enzyme–substrate interactions. The results establish a foundation for rational enzyme engineering aimed at improving catalytic efficiency and substrate tolerance. Such advances could accelerate industrial lignin valorization, supporting the development of bio-based fuels, chemicals, and sustainable materials. Future research should focus on applying these insights to experimental enzyme redesign and integrating them into scalable bioprocesses.

## Figures and Tables

**Figure 1 ijms-26-09378-f001:**
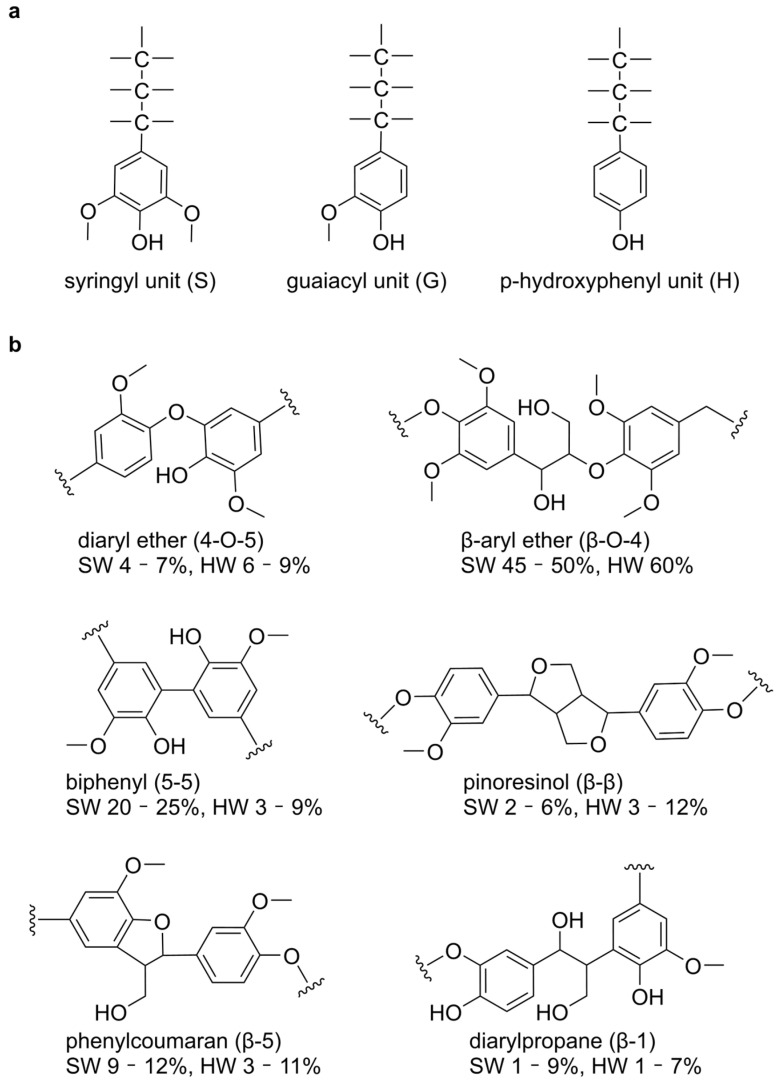
Structural Composition and Bonding Patterns in Lignin Polymers. (**a**) Structure of S, G and H units in lignin polymers. (**b**) Representative linkage types in lignin and their relative abundances in softwoods and hardwoods.

**Figure 2 ijms-26-09378-f002:**
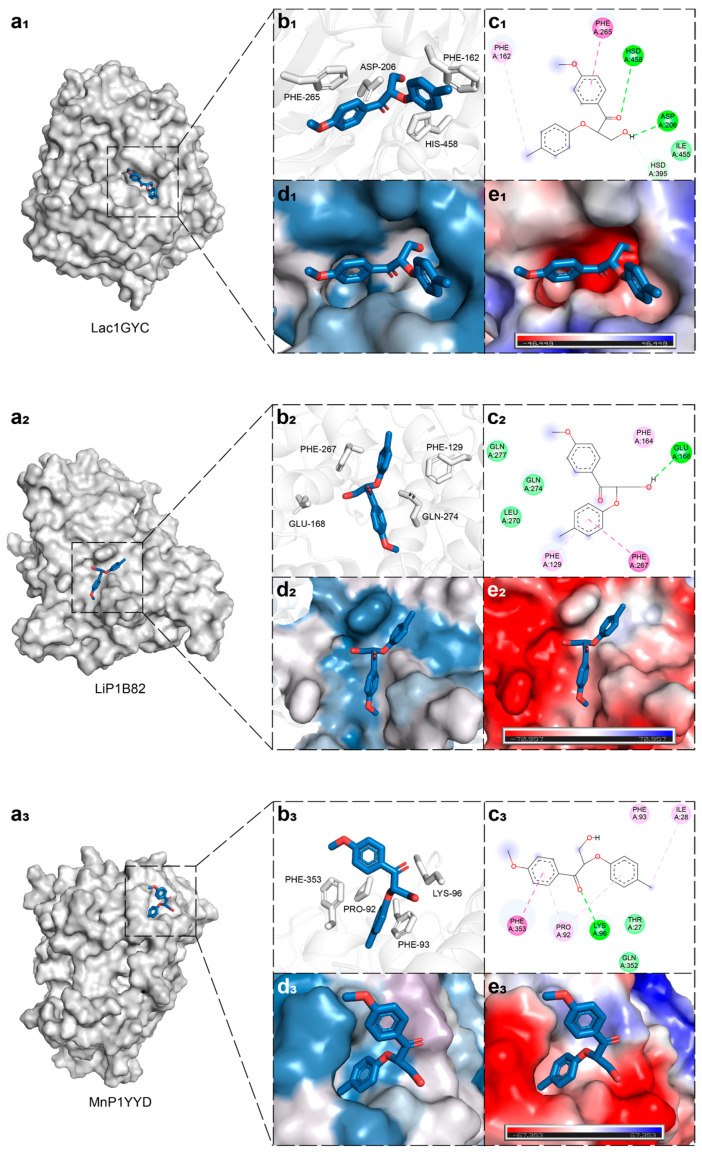
Structural analysis and interaction mapping of the catalytic center in lignin-degrading enzymes. (**a**) Overall structure of the enzyme-substrate complex. (**b**) Close-up view of the substrate-binding site, with key aromatic and polar residues shown as stick models. (**c**) Interaction analysis of the enzyme-substrate docking conformation: light pink residues represent π–alkyl interactions and magenta represents π–π stacking, while light green residues represent van der Waals forces, and light green represents hydrogen bond interactions. (**d**) Hydrophobicity analysis of the active site, where darker blue region indicates stronger hydrophobicity. (**e**) Electrostatic potential surface of the active site, with red denoting negatively charged regions and blue-purple indicating positively charged regions. Subfigures (**a_1_**–**e_1_**,**a_2_**–**e_2_**,**a_3_**–**e_3_**) correspond to Lac1GYC, LiP1B82, and MnP1YYD, respectively.

**Figure 3 ijms-26-09378-f003:**
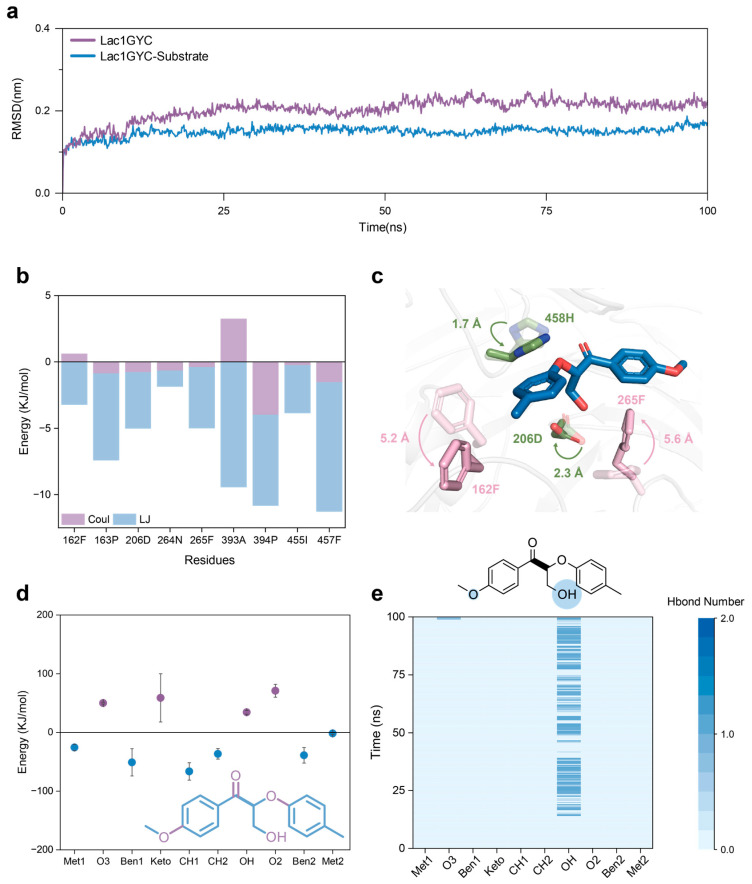
Molecular interactions and structural dynamics of Lac1GYC-substrate complex. (**a**) Time evolution of backbone RMSD with and without substrate. (**b**) Interaction energy between substrate and the laccase. Blue bars represent hydrophobic interactions (LJ, Lennard–Jones potential), while purple bars indicate electrostatic interactions (Coul, Coulombic energy). (**c**) Conformational changes in active site residues. The blue structure represents the substrate lignin dimer, with aromatic residues shown in pink and polar residues in green. (**d**) Calculated binding free energy between the laccase and functional groups of substrates. Purple-labeled groups indicate repulsive interactions, while blue-labeled groups indicate attractive interactions. (**e**) Hydrogen bonding interactions between functional groups of substrate and enzymes. The sizes of the cyan cycles shown in the chemical structures of the substrate represent the strength of the hydrogen bonding interactions.

**Figure 4 ijms-26-09378-f004:**
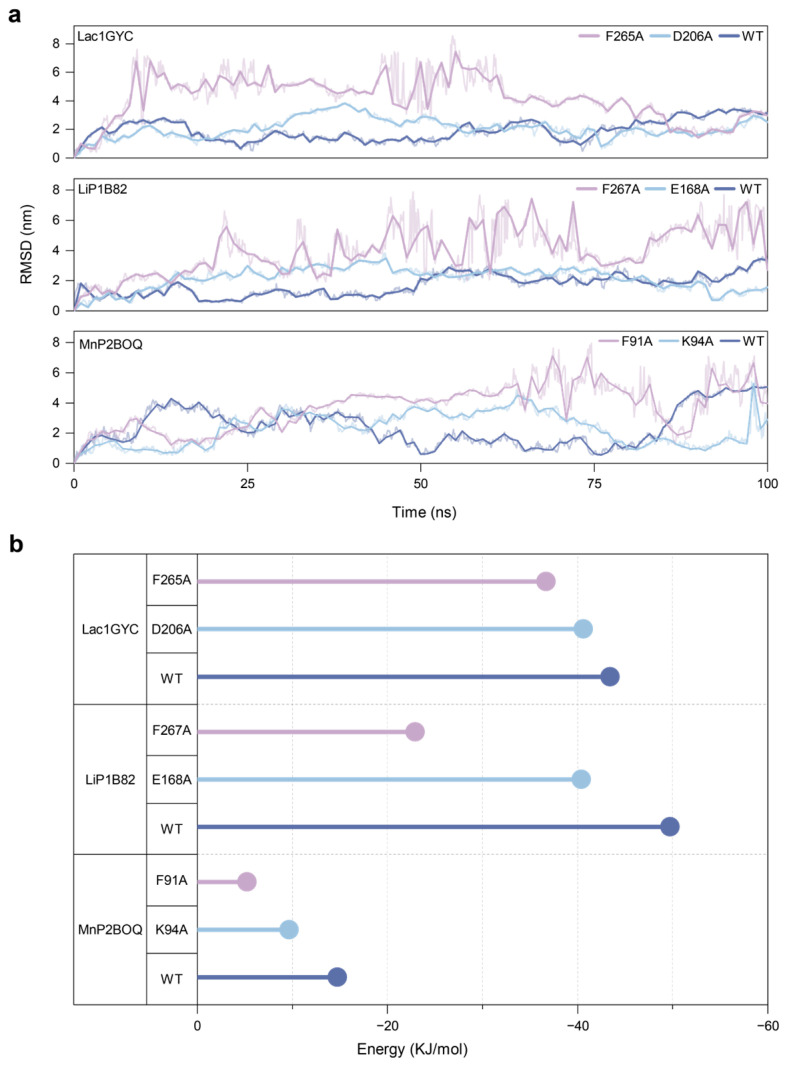
Molecular dynamics analysis of the effects of single-site mutations on enzyme–ligand interaction. (**a**) Time-dependent changes in backbone RMSD for the wild-type and mutant lignin-degrading enzymes. (**b**) Calculated binding free energies of the wild-type and mutant enzymes in complex with the ligand.

**Figure 5 ijms-26-09378-f005:**
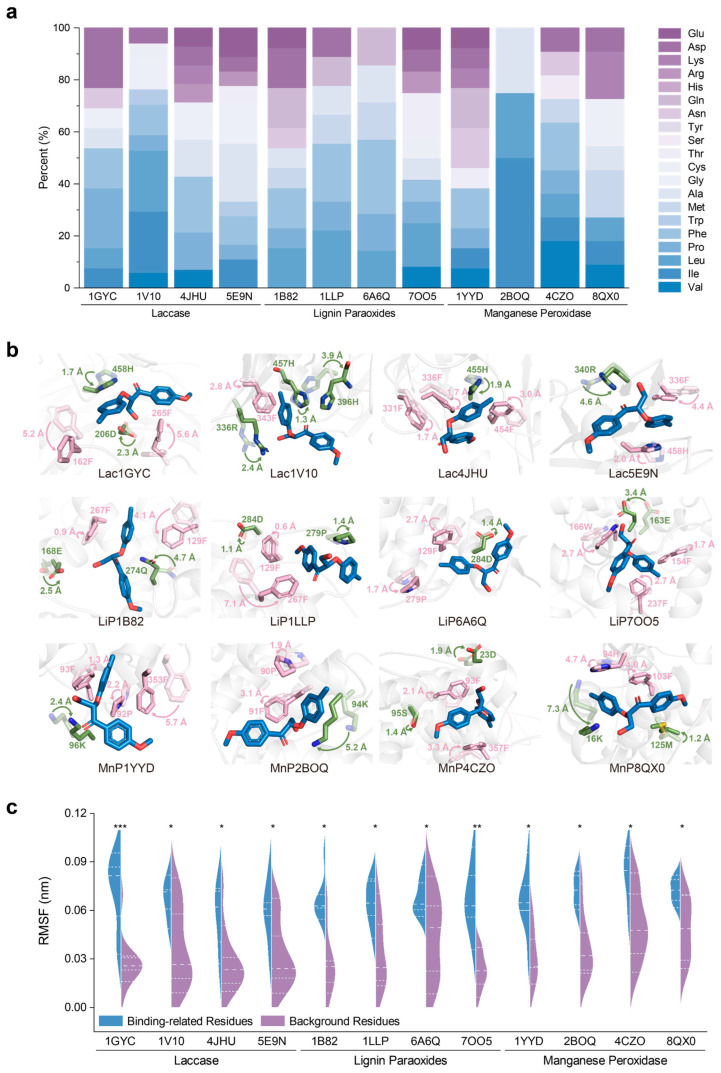
Structural and statistical analysis of binding-related residues in lignin-degrading enzymes. (**a**) Statistical analysis of properties of key amino acids contributing significantly to binding free energy. Darker blue represents higher hydrophobicity, while darker purple indicates increased hydrophilicity. (**b**) Conformational transitions of active site residues before and after MD simulations. Residues involved in hydrophobic interactions are colored pink, while those involved in hydrogen bonding or van der Waals interactions are shown in green. (**c**) Statistical analysis of fluctuations in auxiliary catalytic and binding residues, along with background residues within 5 Å. *p* ≤ 0.05 (*), *p* ≤ 0.01 (**), and *p* ≤ 0.0001 (***) indicate increasing levels of statistical significance.

**Figure 6 ijms-26-09378-f006:**
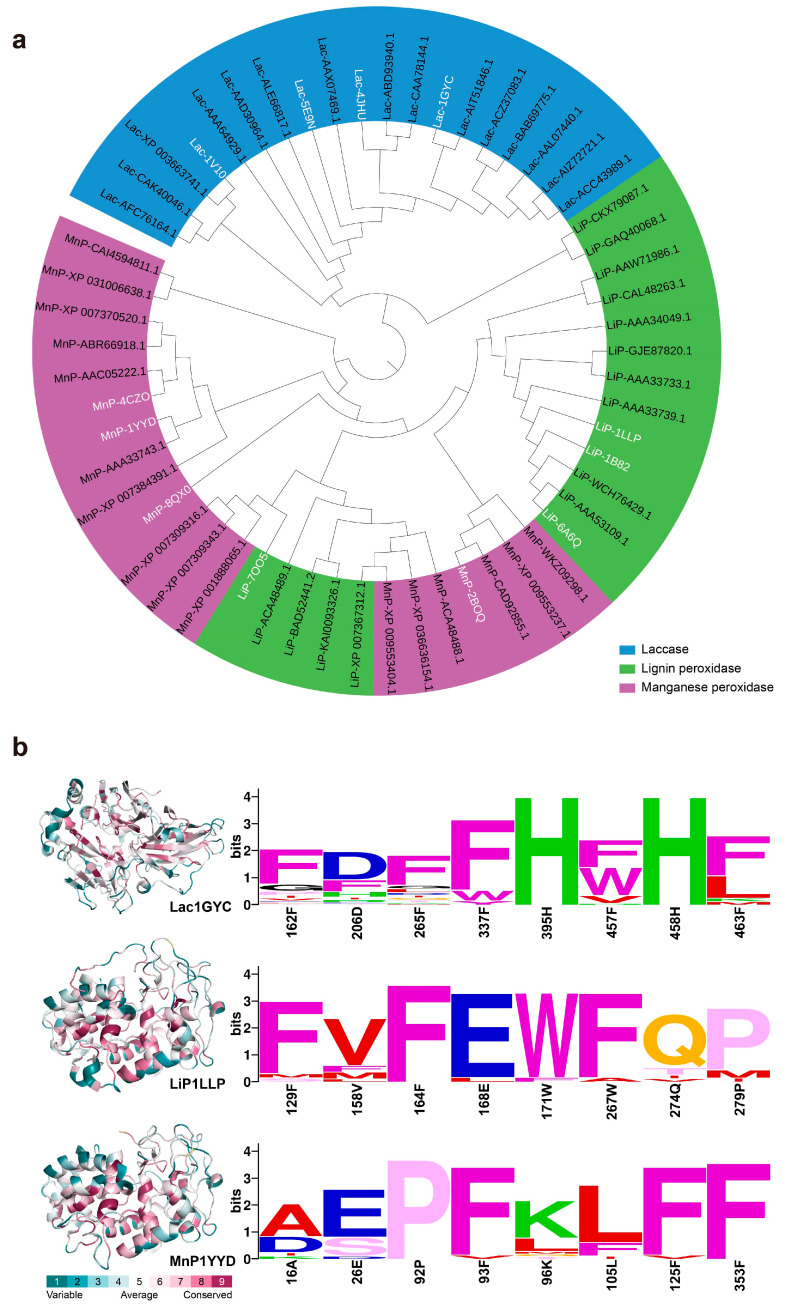
Conservation analysis of lignin-degrading enzymes. (**a**) Phylogenetic tree of 57 lignin-degrading enzymes from NCBI. 19 laccases, 18 lignin peroxidases, and 20 manganese peroxidases are colored blue, green, and purple, respectively. Enzymes highlighted with white labels represent those investigated in this study. (**b**) Sequence conservation analysis. Left panel: overall conservation of lignin-degrading enzymes, with a green-to-red gradient representing increasing conservation. Right panel: conservation of amino acid residues surrounding the active site. Each amino acid is shown as a one-letter code and colored according to physicochemical similarity. In each row of sequence profiles, the heights of all letters represent the relative degrees of conservation, while the height of a single letter denotes its specific occurrence frequency.

## Data Availability

The original contributions presented in this study are included in the article/Appendix A. Further inquiries can be directed to the corresponding author.

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
