# Peer review of "Molecular Dynamics Simulation Reveals the Mechanism of Substrate Recognition by Lignin-Degrading Enzymes"

_ijms, 2025, doi:10.3390/ijms26199378_

Round 1

Reviewer 1 Report

Comments and Suggestions for Authors

This manuscript presents a comprehensive investigation into the substrate recognition mechanisms of laccases, lignin peroxidases, and manganese peroxidases using all-atom molecular dynamics simulations, binding free energy calculations, and virtual mutagenesis. The study identifies a dual recognition mechanism involving hydrogen bonding by polar residues and hydrophobic interactions by aromatic residues, supported by sequence conservation analysis. The topic is scientifically relevant, the methodology is appropriate, and the results are clearly presented with well-prepared figures. Some specific suggestions are listed below.

1. The Introduction includes limited references to recent (past three years) molecular dynamics studies on other lignin-degrading enzymes. Expanding the literature coverage to incorporate the latest relevant work would provide a more comprehensive context for the study.

2. In Figure 3, the meaning of each color code and the statistical methods applied are not fully described. It is recommended to include these details in the figure legend or in the corresponding text to enhance clarity for readers.

3. The Discussion section would benefit from a more explicit comparison between the current findings and previous MD simulation results, including both similarities and differences. Such a comparison would better highlight the novelty and added value of this work.

4. In Figure 5c, significance levels (p-values) are indicated, but the statistical methods used to obtain these values are not specified. Please state the statistical tests performed (e.g., t-test, ANOVA) and the software used.

5. Section 2.5 lacks information on the number of sequences analyzed, the alignment method, and the scoring criteria for conservation. Providing these details would improve methodological transparency and reproducibility.

Author Response

Comments 1: The Introduction includes limited references to recent (past three years) molecular dynamics studies on other lignin-degrading enzymes. Expanding the literature coverage to incorporate the latest relevant work would provide a more comprehensive context for the study.

Response 1: We thank the reviewer for this helpful suggestion. We have added several recent studies (published within the past three years) that utilize molecular dynamics simulations to investigate lignin-degrading enzymes, providing a more comprehensive background and context for the present work. These contexts have been included in revised manuscript (Lines 46-57 on page 2).

Comments 2: In Figure 3, the meaning of each color code and the statistical methods applied are not fully described. It is recommended to include these details in the figure legend or in the corresponding text to enhance clarity for readers.

Response 2: We appreciate this comment. We have revised Figure 3 and its legend to provide explicit explanations for each color code and the statistical methods used, ensuring improved readability for readers. And these changes could be found in the figure legend at lines 168-177 on page 8.

Comments 3: The Discussion section would benefit from a more explicit comparison between the current findings and previous MD simulation results, including both similarities and differences. Such a comparison would better highlight the novelty and added value of this work.

Response 3: We thank the reviewer for this helpful suggestion. We have added several recent studies (published within the past three years) that utilize molecular dynamics simulations to investigate lignin-degrading enzymes, providing a more comprehensive background and context for the present work. These contexts have been included in revised manuscript (Lines 319-323, 328-331 and 337-340 on page 14).

Comments 4: In Figure 5c, significance levels (p-values) are indicated, but the statistical methods used to obtain these values are not specified. Please state the statistical tests performed (e.g., t-test, ANOVA) and the software used.

Response 4: We thank the reviewer for pointing out the missing description of statistical methods in Figure 5c. The significance levels (p-values) in Figure 5c were obtained using the Mann–Whitney U test (two-sided), which is a non-parametric test suitable for comparing two independent groups without assuming normal distribution. For our relatively small sample sizes, we calculated the exact two-tailed p-values by enumerating all possible group allocations and computing the distribution of U statistics, ensuring accurate results without relying on normal approximation. We have now included a detailed description of the statistical test and software in the revised Materials and Methods section. These contexts have been included in revised manuscript (Lines 431-434 on page 16).

Comments 5: Section 2.5 lacks information on the number of sequences analyzed, the alignment method, and the scoring criteria for conservation. Providing these details would improve methodological transparency and reproducibility.

Response 5: Thank you for this valuable suggestion. We have expanded the description of sequence conservation analysis to include the number of sequences used, alignment method, and conservation scoring criteria in the revised “Materials and Methods” section and figure legend of Figure 6b (Lines 297-298 on page 14 and lines 434-436 on page 16).

Reviewer 2 Report

Comments and Suggestions for Authors

This manuscript is clear, well written, concise enough, and contains all the relevant information.   I appreciated the emphasis on the evolutionary convergence, at molecular level, among the active sites of the three classes of ligninolytic enzymes. I just detected some minor typos:

Line 118 What does "enzyme-only complexes"? If the enzymes are alone, why "complexes?

Lines 122-123 Perhaps, something is missing in this sentence. Please check

Figure 3b. What does "coul" mean inside this figure?

Author Response

Comments 1: Line 118 What does "enzyme-only complexes"? If the enzymes are alone, why "complexes?

Response 1: We thank the reviewer for pointing out this imprecise wording. In this study, “enzyme-only complexes” refers to the simulation systems containing only the protein, ions, and water molecules, but without the substrate. We have revised the wording into “enzyme-only systems” for greater clarity.

Comments 2: Lines 122-123 Perhaps, something is missing in this sentence. Please check.

Response 2: We appreciate this suggestion. The original statement was indeed too brief and lacked clarity. We have revised it to include the quantitative indicator of structural stability (RMSD fluctuations) and its implication, thereby improving logical flow. And these changes could be found at line 133-137 on page 7.

Comments 3: Figure 3b. What does "coul" mean inside this figure?

Response 3: We thank the reviewer for the question. “Coul” is the abbreviation for Coulombic interactions, which correspond to the electrostatic interaction energy. We have clarified this in the figure legend. And these changes could be found in the figure legend at line 170 and 171 on page 7.

Reviewer 3 Report

Comments and Suggestions for Authors

It is a perfectly well written manuscript that combines knowledge on enzymes and lignin through the DFT calculations. What I miss in this publication:

(1) Overview what other DFT studies found out for the enzyme-lignin interactions and authors should look globally for articles, not only in china for the Introduction

(2) Discussion does not contain any dialog with the literature. That is very bad and should be corrected with the interaction of at least 5 articles which can help to discuss the results.

(3) Social and economic impact of this study should be mentioned in the Discussion.

(4) Conclusion is missing and should be not longer than 7 sentences with the overall impact of this study on sciences and future research.

Author Response

Comments 1: Overview what other DFT studies found out for the enzyme-lignin interactions and authors should look globally for articles, not only in China for the Introduction

Response 1: We appreciate the reviewer’s recommendation. We have expanded the Introduction to include a more comprehensive overview of relevant DFT studies on enzyme–lignin interactions, with an emphasis on global literature. These contexts have been included in revised manuscript (Lines 46-57 on page 2).

Comments 2: Discussion does not contain any dialog with the literature. That is very bad and should be corrected with the interaction of at least 5 articles which can help to discuss the results.

Response 2: Thank you for this constructive suggestion. We have revised the Discussion section to include explicit comparisons and references to at least five relevant studies, thereby increasing the interpretative depth and scientific credibility of our conclusions. And these changes could be found at lines 307-309, 311-312, 323-325, 328-331, 340-344 and 351-354 on page 8.

Comments 3: Social and economic impact of this study should be mentioned in the Discussion.

Response 3: We agree with the reviewer that emphasizing the potential social and economic impact would enhance the manuscript. We have expanded the Discussion to highlight the relevance of our findings to lignin valorization and biorefinery development. These contexts have been included in revised manuscript (Lines 355-365 on page 15).

Comments 4: Conclusion is missing and should be not longer than 7 sentences with the overall impact of this study on sciences and future research.

Response 4: We appreciate the reviewer’s observation. A concise conclusion section summarizing the main findings, scientific impact, and future research directions has been added in revised manuscript (Lines 366-379 on page 15).

Round 2

Reviewer 3 Report

Comments and Suggestions for Authors

all comments were implemented